# Convergent Evolution of Armor: Thermal Resistance in Deep-Sea Hydrothermal Vent Crustaceans

**DOI:** 10.3390/biology13120956

**Published:** 2024-11-21

**Authors:** Boongho Cho, Sook-Jin Jang, Hee-seung Hwang, Taewon Kim

**Affiliations:** 1Department of Mechanical Engineering, The Hong Kong Polytechnic University, Hung Hom, Kowloon, Hong Kong SAR 999077, China; boonghocho@inha.ac.kr; 2Department of Ocean Sciences, Inha University, 100 Inha-ro, Michuhol-gu, Incheon 22212, Republic of Korea; 3Program in Biomedical Science and Engineering, Inha University, 100 Inha-ro, Michuhol-gu, Incheon 22212, Republic of Korea; 4Ocean Georesources Research Department, Korea Institute of Ocean Science and Technology, Busan 49111, Republic of Korea; jsookjin@kiost.ac.kr; 5BK21 Center for Precision Medicine & Smart Engineering, Inha University, 100 Inha-ro, Michuhol-gu, Incheon 22212, Republic of Korea; 6Scripps Institution of Oceanography, University of California San Diego, La Jolla, CA 92093, USA; heeseung0212@gmail.com

**Keywords:** extremophile, crustacean, exoskeleton, thermal stability

## Abstract

In 1977, deep-sea hydrothermal vents, which are high-temperature, high-pressure environments, were discovered by humans, and how the creatures there adapted to their habitats is still being unveiled. We are addressing the question of whether there are specific adaptations in the exoskeletons of crustaceans inhabiting deep-sea hydrothermal vents that enable them to withstand high temperature and pressure conditions. We observed that two vent crustaceans, through convergent evolution, possess thermal stability that is 2.8 times higher than that of four different coastal species, allowing them to survive in the extremely high-temperature environments of their habitat. This is influenced by different compounds with similar elemental composition. The insights gained from studying crustaceans in such extreme environments can offer invaluable insights into the ongoing advancements of biological evolution.

## 1. Introduction

On Earth, there are extreme environments such as polar regions, deep seas, hydrothermal vents, deserts, and volcanoes, where it is believed to be difficult for life to survive. Each extreme environment on Earth is exposed to its own unique abiotic factors: Antarctica experiences ultralow surface temperatures reaching −98 °C [1], the deep sea faces high pressure due to depths exceeding 10,000 m [2] and temperatures close to 0 °C [3], while deep-sea hydrothermal vents are subjected to high pressure because of an average depth exceeding 2100 m and extreme heat with vent fluid temperatures averaging 400 °C [4]. However, some organisms, known as extremophiles, are able to survive in these extreme environments (e.g., temperature, salinity, and pH) [5,6]. Research on these organisms has been very limited to date, as their habitats do not easily allow human access, and their remarkable vitality is expected to have high research value [7]. Research of animals from vents typically employs specialized tools and techniques, such as remotely operated vehicles or television grab for sampling, as well as high-pressure incubation systems, genetic and mechanic analysis methods, to investigate their unique features to extreme environments.

Hydrothermal vents are one of the most extreme environments on Earth, and the temperature and pressure are higher than those in other areas. Extremophiles, such as snails and archaeon living in hydrothermal vents have incomparable survival mechanisms. These include the strong iron-plated armor covering snails [8] and unidentified mysterious metabolism of the thermoacidophilic archaeon that can endure low pH and high temperature [9]. Recent studies reported that the exoskeletons of crabs inhabiting deep-sea hydrothermal vents have good mechanical properties (hardness and reduced modulus) [10], and notable thermal stability [10,11] compared to coastal crabs. The structure and composition of their exoskeletons are believed to contribute to these properties. Consequently, the evolution of crustaceans is closely linked to the characteristics of their exoskeleton [12]. However, to date, it remains unclear whether these traits are unique to specific species or represent a convergent characteristic among crustaceans inhabiting deep-sea hydrothermal vents. Similarly, while numerous studies focused on the unique survival mechanisms of individual species, research on convergent evolution—where genetically unrelated species in the same habitat develop identical survival strategies—remains scarce.

Here, exoskeletons of two vent crustaceans were compared with those of four coastal crustaceans to determine if the deep-sea hydrothermal vent crustaceans evolved extraordinary features of exoskeletons to endure high temperature and pressures. In previous research, the vent species *Austinograea* sp. was shown to have an exoskeleton highly resistant to temperature and pressure [10]. It suggests that the exoskeletons of hydrothermal vent crustaceans may have evolved similarly under the same selection pressure. Thus, we hypothesized that the exoskeleton of crustaceans inhabiting the vent would have undergone convergent evolution to withstand the extreme environments of high temperatures and pressures. Accordingly, we predicted that the thermal stability and mechanical properties of the exoskeleton would be higher in vent crustaceans than those in coastal habitats.

## 2. Materials and Methods

### 2.1. Sample Preparation

Two species of vent crustaceans [*Austinograea* sp. (vent crab) and *Munidopsis lauensis* (vent squat lobster)] and four species of coastal crustaceans [*Charybdis japonica* (Asian paddle crab), *Portunus trituberculatus* (blue crab), *Elassochirus cavimanus* (purple hermit crab), and *Oratosquilla oratoria* (mantis shrimp)] were used for this study (Figure 1a and Appendix A and Table 1). The vent species were collected from the Onnuri vent field (OVF) of the Indian Ocean using TV grab (Oktopus GmbH, Hohenwestedt, Germany) installed on the ISABU research vessel.

Although there are clear differences in the zonation of organisms at each hydrothermal vent [14], crustaceans were located closest to the vent sources in the OVF, which is similar to the East Scotia Rise (Southern Ocean) hydrothermal area [15]. Through observations utilizing a remotely operated vehicle Ropos (Canadian Scientific Submersible Facility, North Saanich, BC, Canada) and a TV grab camera, it has been shown that certain crustaceans exhibit a behavior of traversing across hydrothermal vents, indicating robust to high temperatures. The research area, OVF, is a diffusion vent unlike typical chimney-dominated hydrothermal regions [16], so crossing over the vent where the flume emerges is very similar to crossing a general flat terrain. To ensure that these analyses were not specific to a certain taxon, we selected species from Brachyura and Anomura that inhabit different habitats (Table 1).

Samples with similar size, which were measured using a caliper (CD-15PSX, Mitutoyo, Kanagawa, Japan), were used (Table 1). All samples were stored immediately in 75% ethanol after collection. A single 5 × 5 mm square sample from the flattest carapace (mesogastric region) of the exoskeleton was used for mechanical, structural, and compositional analysis [11]. Thermal stability analysis was performed on the nearby regions, including potogastric, urogastric, and epigastric. The samples for a nanoindenter, SEM, EDX, and Raman spectroscopy were embedded in cold resin to avoid temperature-induced denaturation. Furthermore, they were finely polished to a 5 μm using an auto polisher (5-2600, Allied High Tech Products, Inc., Compton, CA, USA). The impurities in the TGA samples were removed by passing them through a 500 μm mesh sieve. The fracturing method was used for microstructure analysis of the exoskeleton [17].

### 2.2. Property Analysis

The overall analysis consisted of four steps: structural, component, mechanical, and thermal analyses. Scanning electron microscopy (SEM: S-4300SE, Hitachi, Ltd., Tokyo, Japan) was used for structural analysis, and the thickness of each layer was measured using ImageJ 1.54g^®^. The elemental and compound analysis of each layer constituting the exoskeleton was performed by energy-dispersive X-ray spectroscopy (EDX; S-4300SE, Hitachi, Ltd., Tokyo, Japan) and Raman spectroscopy (LabRAm HR Evolution, HORIBA, Ltd., Kyoto, Japan; laser line: 532 nm, acquisition time (s): 3, accumulation: 10–15, ND Filter (%): 10–50), respectively. The coating machine (Q150T, Quorum technologies Ltd., Ashford, UK) was used for SEM and EDX analyses. Platinum was applied as the coating material, with each coating session lasting 120 and 20 s for each SEM and EDX analysis.

For mechanical properties analysis, the hardness and reduced modulus of each exoskeleton layer were measured using a nanoindenter [(G200, KLA, Milpitas, CA, USA; force: 0.081 gf, time to load: 30 s, peak hold time: 10 s, tip: XP Berkovich diamond tip (20 nm radius)]. The results were calculated by taking the maximum of three measurements obtained from each of the four layers composing each individual, and then repeating this process three times and averaging the results, but the epicuticle layer was very thin, approximately 1% of the exoskeleton, so only one maximum value was used. The mechanical properties can vary depending on the dryness level of the sample, and this level can vary depending on the species [18,19,20]. We ensured that all samples were adequately dehydrated before proceeding with the comparative analysis, aligning with the approach taken in many previous studies that also utilized dehydrated samples [18,21,22]. To control the dehydration level of the samples stored in ethanol, they were stored for at least one month. This duration is adequate for complete dehydration [23,24].

Thermal analysis was conducted by thermogravimetric analysis (TGA; STA 409 PC, NETZSCH, Selb, Deutschland; heating rate: 10 °C/min, atmosphere: nitrogen gas) from room temperature to 800 °C, where calcium carbonate was combusted. The main combustion stages were composed of three ranges: Range 1 (30–200 °C) [25,26], Range 2 (250–500 °C) [25,27], and Range 3 (600–800 °C) [28,29]. In Ranges 1 through 3, the representative compounds that typically combust are volatile substances, organic substances, and inorganic substances, respectively [10]. The thermal stability evaluation compared the weight loss (%) of the exoskeleton after combustion; a lower weight loss (%) indicated higher thermal stability [30]. Range 1 was the evaluation temperature range for testing the thermal stability because the initial temperature at which the vent fluid mixes with the surrounding cold seawater is approximately 230 °C [31] and the main combustion temperature of the exoskeleton before 230 °C was up to approximately 200 °C.

### 2.3. Statistical Analysis

One-way ANOVA for normally distributed data and a Kruskal–Wallis H test were conducted for non-normally distributed data to compare the crustacean exoskeleton characteristics depending on the habitat and infraorder Brachyura. If there were significant differences, Scheffé’s test was used as a post hoc test. A two-tailed independent *t* test was conducted to identify the difference between two Anomura vent squat lobsters and purple hermit crabs. All statistical analyses were performed using SPSS software (version 19.0; SPSS, Inc., Chicago, IL, USA). All data were presented as the mean ± standard error (SE).

Phylogenetic generalized least squares (PGLS) in R (version 4.3.3) were conducted to understand correlations between exoskeletal characteristics and habitats. We used the ape package for phylogenetic analysis and the nlme package for linear model fitting with a Brownian motion correlation structure (corBrownian).

### 2.4. DNA Extraction and Molecular Analysis

Genomic DNAs of specimens were extracted from the muscle tissue of a pereiopod using QIAamp Fast DNA Tissue Kit (Qiagen, Inc., Hilden, Germany), following the manufacturer’s protocol. The partial fragments of two mitochondrial genes, cytochrome c oxidase (*mtCOI*) and 16S ribosomal RNA (*mt16S rRNA*), and two nuclear genes, *18S* ribosomal DNA (*18S rRNA*) and histone-3 (*H3*), were amplified using previously published primers (Appendix A). Polymerase chain reaction (PCR) was conducted using IP-Taq Master mix (Cosmogenetech, Seoul, Republic of Korea) and the thermocycling followed same cycles with specific annealing temperature to the primer sets (Appendix A). PCR products were sequenced using an ABI 3730xl Analyzer (Applied Biosystems, Foster City, CA, USA) and modified manually using Geneious Prime 2022.2. Sequences newly generated in this study were registered in GenBank under the accession IDs: OQ644532–OQ644535, OQ644545, OR462783–OR462785 for *mtCOI*; OQ629571–OQ629584, OR467411–OR467412 for *mt16S rRNA*; OQ629560, OQ629561, OR467415 for *18S* rRNA; and OQ629549–OQ629556, OR508504–OR509505 for *H3*.

Phylogenetic analysis was conducted using concatenated sequence data (2101 bp) of *mtCOI*, *mt16S rRNA*, *18S rRNA*, and *H3* genes, and mantis shrimp was utilized as an outgroup based on the previously known systematics (Appendix A). The *mt*COI and 18rRNA sequences of the vent crab, Asian paddle crab, vent squat lobster, and purple hermit crab were obtained through previous study [13].

The partitioned data set, with applied specific substitution models for each gene was, used for both Bayesian inference (BI) and Maximum likelihood (ML) analysis (Appendix A). BI and ML approaches were applied to infer phylogenetic relationships, for which MrBayes 3.2.7 [32,33] and IQ-Tree v. 2.2.2.7 [34,35] on the CIPRES web server [36] were performed, respectively (Appendix A).

## 3. Results

### 3.1. Structural Characteristics

All six crustacean exoskeletons are comprised of four layers: epicuticle, exocuticle, endocuticle, and membrane. However, the exoskeleton of the purple hermit crab lacked a membrane layer. The epicuticle layer was composed of a granular structure, and the exocuticle and endocuticle were made with a Bouligand structure. The membrane had a multilayer structure (Appendix A). A significant difference in thickness ratio (%) was observed in each layer among species (Kruskal–Wallis *H* test; epicuticle: χ^2^ = 16.131, d.f. = 5, *p* = 0.006, exocuticle: χ^2^ = 15.379, d.f. = 5, *p* = 0.009, endocuticle: χ^2^ = 15.919, d.f. = 5, *p* = 0.007, and membrane: χ^2^ = 10.718, d.f. = 4, *p* = 0.03), but there was no significant difference between the two habitats (Figure 2c).

### 3.2. Characteristics of Components

The exoskeletons of vent squat lobsters, vent crabs, and Asian paddle crabs consisted of twelve elements (C, N, O, Na, Mg, Al, Si, P, Zr, S, Cl, and Ca). The exoskeletons of blue crabs, purple hermit crabs, and mantis shrimps had iron (Fe) in addition to the twelve elements (Appendix A). There were no differences between the elements in each layer of the exoskeleton across the two habitats for the six species (Appendix A). Considering the kind of exoskeleton compounds, those in hydrothermal crustaceans include calcite, calcium phosphate, protein, C-O-C stretch, α-chitin, and H_2_O. In contrast, coastal crustaceans contained carotenoid-based compounds in addition to the components of the hydrothermal crustacean compounds (Figure 3 and Appendix A and Appendix A).

### 3.3. Thermal Stability

In the three temperature ranges (Range 1, Range 2, and Range 3), there was a significant difference in thermal stability among the six species (one-way ANOVA; Range 1: *F*_5,18_ = 290.141, *p* < 0.001, Range 2: *F*_5,18_ = 33.398, *p* < 0.001, Kruskal–Wallis *H* test; and Range 3: χ^2^ = 17.358, d.f. = 5, *p* = 0.004; Figure 4a and Appendix A). In Range 1, the thermal stability of the two hydrothermal species was higher than that of the coastal species, but in Range 3, it was the opposite (Scheffé’s post hoc test, *p* < 0.05; Figure 4b). There were significant differences between thermal stability of exoskeletons and habitats when using the regression coefficient of the phylogenetic relationship (PGLS; Range 1: slope: 6.877, *T* value: 3.315, *p* = 0.03; Range 3: slope: −9.924, *T* value: −3.136, *p* = 0.035).

### 3.4. Mechanical Properties

Significant differences in mechanical properties were observed at the epicuticle and exocuticle (Kruskal–Wallis *H* test; epicuticle hardness: χ^2^ = 13.913, d.f. = 5, *p* = 0.016, one-way ANOVA; epicuticle reduced modulus: *F*_5,18_ = 51.003, *p* < 0.001, exocuticle hardness: *F*_5,18_ = 15.613, *p* < 0.001, and exocuticle reduced modulus: *F*_5,18_ = 30.927, *p* < 0.001), but there was no significant difference between the two habitats (Figure 2a,b).

When comparing mechanical properties within infraorders, Brachyura showed superior hardness in the exocuticle than coastal species (one-way ANOVA: *F*_2,8_ = 15.174, *p* = 0.004), while Anomura exhibited superior hardness and reduced modulus in both epicuticle and exocuticle compared to coastal species (*t* test; epicuticle hardness: *t* = 11.240, *N*_1_ = 4, *N*_2_ = 3, *p* = 0.013, epicuticle reduced modulus: *t* = 6.927, *N*_1_ = 4, *N*_2_ = 3, *p* < 0.001, exocuticle hardness: *t* = 3.609, *N*_1_ = 4, *N*_2_ = 3, *p* = 0.01, and exocuticle reduced modulus: *t* = 1.679, *N*_1_ = 4, *N*_2_ = 3, *p* < 0.01).

### 3.5. Phylogenetic Tree

The phylogenetic reconstructions based on the Bayesian inference and maximum-likelihood approaches exhibited a well-supported divergence between Brachyura and Anomura (Figure 1b). Within brachyuran crabs, the vent crab diverged earlier from the ancestral lineage.

## 4. Discussion

Comparative analysis provided evidence for the convergent evolution of exoskeletons in crustaceans inhabiting deep-sea hydrothermal vents. The vent squat lobster and vent crab evolved independently, but the thermal stability of the two crustaceans was higher than that of the four species in the coastal habitats. On the other hand, the mechanical properties of the exoskeleton did not differ significantly between the two habitats. Our study provides evidence that crustaceans living in hydrothermal vents commonly exhibit resistance to high temperature.

Among the three combustion ranges in the thermal stability analysis, significant differences were found between the two habitats in Ranges 1 and 3. In Range 1 of thermal stability analysis, water or volatile substances constituting chitin [25,26,27,30] and astaxanthin (ATX) [37] were combusted. The weight loss (%) of the vent species was only 36% of that of the coastal ones. The water content of the chitin isolated from the exoskeleton was approximately 5–6% [38,39]. Thus, the vent species combusted only substances from chitin, but the coastal species contained compounds with a water content, such as ATX. We evaluated the thermal stability of the exoskeleton within a specific temperature range: Range 1. The likelihood of combustion of water within the exoskeleton was significantly higher in deep-sea conditions, where there is an abundance of water compared to air. This is due to water’s higher thermal conductivity, resulting from its greater molecular density. For reference, water’s thermal conductivity is approximately 0.6 W/m∙K [(provided by the NIST; www.nist.gov (accessed on 10 June 2020)] [40], whereas air’s thermal conductivity is much lower, at less than 0.03 W/m∙K [41,42]. Consequently, the thermal effect on the crustacean exoskeleton is expected to be more significant in water than in air. Additionally, under conditions similar to the deep-sea hydrothermal vent environments, where the depth over 2000 m (<200 bar), the thermal conductivity of water (W/m °C) increases as the temperature rises until approximately 200 °C [40]. As a result, vent species may experience more heat stress due to high pressure compared to coastal species. The weight loss (%) in Range 2 was attributed to chitin decomposition, the main organic compound, and other organic compounds constituting the exoskeleton [25,27,28]. The organic compound of the vent exoskeleton was only chitin, as confirmed by Raman spectroscopy. On the other hand, coastal species also had carotenoid-based compounds (ATX; degradation range: 250–450 °C) [43] known as red pigment, and unsaturated fatty acids (degradation range: 220–365 °C) [44]. The carotenoid-based compounds are obtained from the crustacean’s diet [45]. Their absence in vent species might be due to environmental and ecological factors. The absence of light in the habitat is one of the major factors. The main uses of pigments in animals is for reproduction [46], camouflage from predators [47], and communication [48]. Vent species frequently have a white color known as albification [49] because they do not need these in dark environments. Furthermore, the crustaceans in the two habitats differ in their ecological niches. The vent species do not need camouflage as they are the top predators in their habitat [50], whereas coastal species are mainly secondary consumers whose camouflage is essential to escape predators [51]. The pigment may be related with the high proportion of calcium carbonate in the vent species. The weight loss (%) in Range 3 was attributed to the decomposition of CaCO_3_, which is the main inorganic compound [28,29,30]. The weight loss in Range 3 of the vent species was 1.5 times higher than that of the coastal species. One notable characteristic of CaCO_3_ is its excellent thermal stability, making it valuable in various fields, such as in biomarker [52] and bioinspired materials [53]. Unlike coastal species, vent species do not have a proportion of related pigment components in the exoskeleton, so the proportion of other components may have increased. Second, the calcium concentration of the sediments near the vents is higher than that at the coast and may have affected the CaCO_3_ content in the exoskeleton. Calcium in marine sediments is supplied mainly from hydrothermal fluids, CaCO_3_ precipitation, and dissolution [54,55]. For example, the calcium concentration in the sediment of the OVF is about 50–84% [56], but the coastal habitats, such as the Yellow and East China Seas, are less than 30% [57].

Previously, heat resistance, within a temperature variation of less than 50 °C, of organisms was predominantly understood as phenotypic plasticity [58,59]. However, our study expands upon this perspective by examining the extreme conditions of deep-sea hydrothermal vents, characterized by a wide temperature variation ranging from 0 to 200 °C. The vent crustaceans under investigation may have pursued a distinct adaptive pathway, evolving exoskeletons to directly counter the thermal stress prevalent in their habitat. This suggests a departure from reliance on phenotypic plasticity within a specific taxon, indicating a more specialized evolutionary strategy crafted to address the unique challenges posed by the deep-sea hydrothermal vent environment.

The main features affecting the mechanical properties of a material are its structure [60], thickness [11,61], and components [10,11,62]. The structure of each exoskeleton layer was the same in all four species, and there were no significant regional differences in thickness and component. A previous study reporting comparative analyses showed that vent crabs had superior mechanical properties and thermal stability to Asian paddle crabs [10]. On the other hand, in this study, there was no difference in mechanical properties between habitats when investigating species within each habitat. However, there are notable differences between the infraorder in different habitats. The vent species exhibited a similar direction of adaptation, which involved strengthening their exoskeleton which protects themselves from high pressure compared to coastal species. Nonetheless, the degree of this strengthening was relatively weaker, providing less conclusive evidence for convergent evolution.

The composition is essential for determining the characteristics of the exoskeleton [8,10,11,63], which are strongly influenced by the sediment [63], seawater [63], and diet [45] near the habitat. In detail, the vent sediments are unique because of the high concentrations of Fe, Si, Ba, Cu, and Zn, originating from the hydrothermal fluid, and S and Mg from seawater [56]. The fluid from the hydrothermal vent interacts with the surrounding rock layers, dissolving the minerals. Those compounds are discharged into seawater, altering their composition [64]. Additionally, because of the absence of sunlight at extreme depths, the prey organisms available to vent crustaceans lack carotenoid-based compounds. Nevertheless, the ratio and diversity of the exoskeleton elements of vent crustaceans were similar with those of coastal crustaceans. In other words, the composition of the elements that made up exoskeleton is similar, but their compound is rather different.

According to PGLS analysis, vent crustaceans, in contrast to coastal crustaceans, possess exoskeletons with enhanced thermal stability. Phylogenetic inferences indicate that the vent squat lobsters and vent crabs have undergone independent evolutionary processes after divergence. The infraorder Anomura, including vent squat lobster, diverged 259 Mya from the crustaceans of Decapoda according to previous phylogenetic estimation [65]. Subsequently, the vent crab (bythograeid crabs in the hydrothermal vent) diverged from their sister taxa about 150–170 Mya, as estimated from the mitogenome sequences [66]. Given the phylogenetical relationship among the taxa, this study strongly supports that even phylogenetically distant species can evolve convergently to have the same thermal and compound properties to adapt to the extreme environment.

Due to the increased pressure at depths over 2000 m in vent environments and the elevated temperatures caused by high-temperature vent fluids (~400 °C), organisms in such environments might experience greater pressure [67]. Therefore, vent organisms likely require mechanisms to withstand these high pressures. It is supposed that crustaceans have a relevant mechanism related to the mechanical properties of their exoskeleton.

Convergent evolution driven by environmental factors is well known across nature and has been studied through various methods, including morphological and genomic analyses. For example, the fossil of an ancient mosasaur, aquatic tetrapod, reveals convergent evolution through preserved morphological traits, such as soft tissues and anatomical details. From a genomic perspective, the photosymbiotic bivalve *Fragum sueziense* showed evidence of parallel or convergent evolution in dark conditions, exhibiting molecular mechanisms similar to those seen in animal–algal photosymbiosis in distantly related lineages such as cnidarians [68]. Similarly, pandas from different family exhibit convergent traits, such as bamboo diet and adaptive pseudothumb, which have been investigated through genomescale analyses [69]. Phenomic data derived from natural history collections and comparative genomics have been suggested as valuable approaches for interpreting convergent evolution [70]. This study employed both morphological traits and genomic data to evaluate convergent evolution; however, additional specific genomic analyses could provide deeper insights.

## 5. Conclusions

The key implication of this study is to reveal differences in the characteristics of the crustacean exoskeleton depending on the unique abiotic factors in habitats and to interpret its evolutionary aspect. Crustaceans in hydrothermal vents may have undergone convergent evolution that improves the thermal stability of their exoskeletons to survive at high temperatures. This suggests that the exclusively shared characteristics among vent crustaceans are adaptations of those animals to extreme environments such as high temperatures.

## Figures and Tables

**Figure 1 biology-13-00956-f001:**
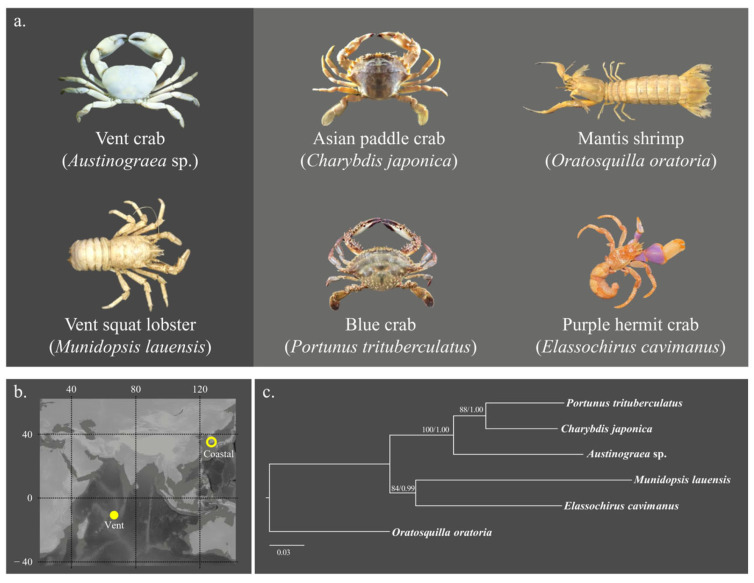
Research was conducted on species (**a**), sampling sites (**b**), and phylogenetic relationships (**c**). The phylogenetic relationship was estimated using two specimens of each species based on BI and ML analysis using the combined data set (2101 bp) of *mtCOI*, *mt16S rRNA*, *18S rRNA*, and *H3* genes. The number at each node represents the bootstrap values from ML analysis, and the posterior probabilities from BI. The scale bar indicates phylogenetic distance of 0.03 nucleotide substitutions per site.

**Figure 2 biology-13-00956-f002:**
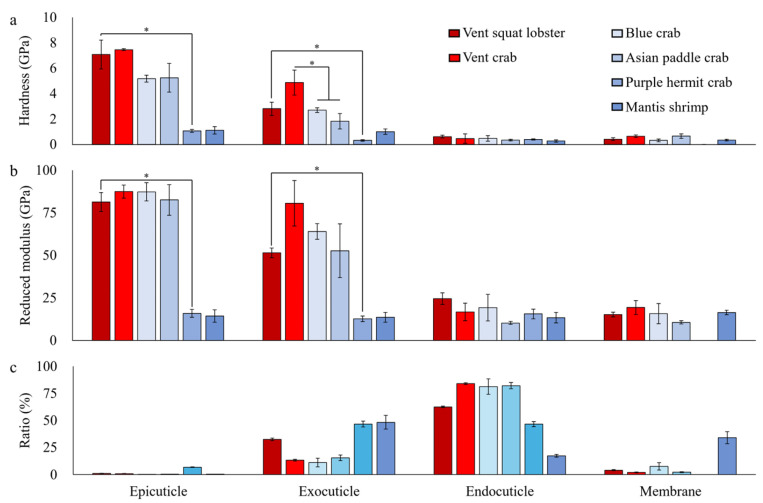
Mechanical properties and thickness ratio of each layer of the exoskeleton. (**a**) hardness, (**b**) reduced modulus, and (**c**) thickness ratio; mean ± SE; the significant difference is indicated by an asterisk (*).

**Figure 3 biology-13-00956-f003:**
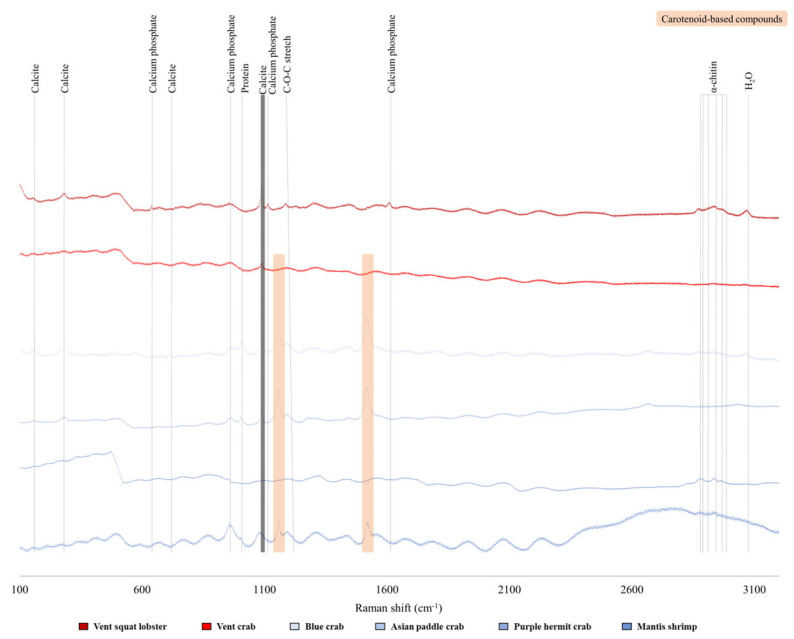
Crustacean exoskeleton (endocuticle layer) compound analysis graph through Raman analysis.

**Figure 4 biology-13-00956-f004:**
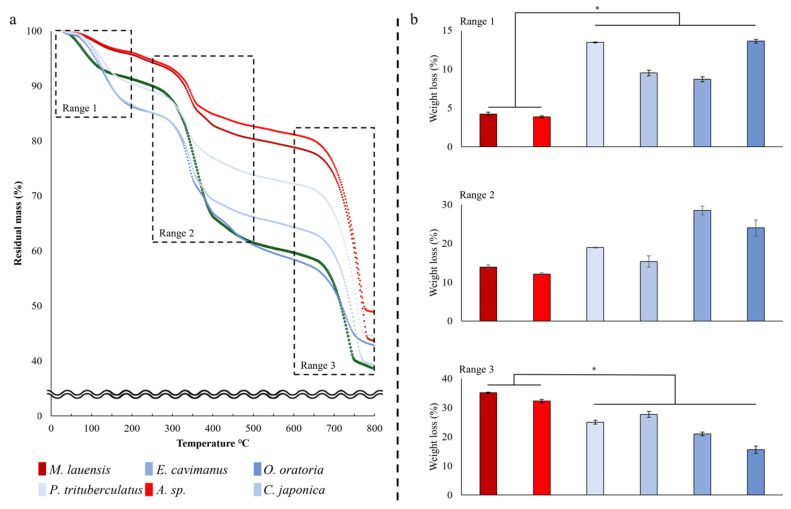
Thermal stability analysis results. (**a**) TGA curve of crustaceans and each combusting substance on each range. Weight loss (%) for each temperature range: (**b**) Range 1, Range 2, and Range 3; mean ± SE; the significant difference is indicated by an asterisk (*).

**Table 1 biology-13-00956-t001:** Detail sample information.

Infraorder	Species	Lat.		Long.		Date (yyyy.mm.dd)	Depth (m)	N ^1^	Size ^2^ (mm)	Habitat	Ref.
Anomura	*Munidopsis lauensis*	11.24	S	66.25	E	2019.06.28	2014	3	25.16 ± 0.24	Vent	[13]
		11.24	S	66.25	E	2019.07.01	2023	1		Vent	This study
Brachyura	*Austinograea* sp.	11.24	S	66.25	E	2019.06.29	2014	3	3.42 ± 0.02	Vent	[10]
Anomura	*Elassochirus cavimanus*	36.20	N	129.41	E	2022.09.01	100	3	18.45 ± 1.56	Coastal	[13]
Brachyura	*Charybdis japonica*	37.45	N	126.60	E	2020.04.06	<50	3	5.74 ± 0.03	Coastal	[10]
Brachyura	*Portunus trituberculatus*	37.45	N	126.60	E	2021.06.21	<50	3	126.31 ± 2.98	Coastal	This study
-	*Oratosquilla oratoria*	34.61	N	127.72	E	2020.06.30	<50	3	171.5 ± 0.75	Coastal	This study

^1^ Sample size; ^2^ carapace width.

## Data Availability

The raw data supporting the conclusions of this article will be made available by the authors on request.

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
