# Peer review of "Convergent Evolution of Armor: Thermal Resistance in Deep-Sea Hydrothermal Vent Crustaceans"

_biology, 2024, doi:10.3390/biology13120956_

Round 1

Reviewer 1 Report

Comments and Suggestions for Authors

Review for the paper “Convergent Evolution of Armor: Thermal Resistance in Deep Sea Hydrothermal Vent Crustaceans” by Boongho Cho and co-authors submitted to “Biology”.

The authors of this research paper conducted an analysis of the exoskeletal adaptations exhibited by crustaceans inhabiting extreme environments, specifically deep-sea hydrothermal vents. They compared the exoskeletons of two species from these extreme habitats with those of four coastal species. They found that the exoskeletons of the vent-dwelling crustaceans exhibited significantly enhanced thermal stability compared to their coastal counterparts. The results indicated that the exoskeletons of these vent crustaceans had a reduced proportion of volatile components, such as water, and an increased proportion of calcium carbonate. These compositional differences suggest a functional advantage in terms of structural integrity and resistance to the extreme conditions found near hydrothermal vents. Additionally, the study noted the absence of carotenoid pigments in the exoskeletons of vent crustaceans, as these pigments are known to have low heat resistance, further supporting the notion of adaptation to extreme temperatures. The authors did not find significant differences in the mechanical properties of the exoskeletons between vent and coastal species. This aspect suggests that while thermal stability has been enhanced, the general mechanical resilience of the exoskeletons remains consistent across both environments. The results of this study may have important implications for our understanding of evolutionary processes in extreme environments. They contribute to the broader discourse on how organisms adapt morphologically and chemically to survive in harsh conditions.

The authors used standard methods and approaches in their study, and only minor revisions are required to improve the paper.

Abstract.

Introduction

L 60. The authors should provide information about the tools and techniques used in studying animals from hydrothermal vents.

L 62. The authors should define the term extremophiles and you provide specific examples of organisms that thrive in extreme environments.

L 67. The authors should mention the specific mechanical properties of the exoskeletons that have been found to be "good" compared to those of coastal crabs.

Materials and Methods.

Figure 1. It is difficult to understand the location of the study areas as they overlap with photographs of crustaceans. The authors should merge this figure into several plates for a better presentation. They should include a coordinate grid with the map.

L 174. The authors should describe in detail the specific protocols that were followed during the extraction of genomic DNA using the QIAamp Fast DNA Tissue Kit.

The authors should specify what size is assumed for each species from Table 1.

Section 2.5 is redundant and should be deleted.

Results.

L 204-206. The authors should delete this text.

Figure 2. The authors should increase the font size.

Section 3.4. In terms of statistical analysis, how do you ensure that the sample sizes (N1 and N2) used in t-tests are adequate to support the conclusions drawn from the study?

Discussion.

L 268. The authors should provide some hypotheses on how the convergent evolution observed in vent squat lobsters and vent crabs reflects their ecological needs in deep-sea habitats.

L 308. The authors should discuss in more detail how the greater weight loss attributed to CaCO3 decomposition in vent species correlates with their overall physiological resilience under high-stress conditions.

L 333. What mechanisms might underlie the observed lack of significant differences in mechanical properties between habitats despite different environmental stresses?

L 340. The authors should specify how differences in the diet of vent crustaceans compared to coastal species might affect the elemental composition of their exoskeletons.

Specific remarks.

L 54. Consider replacing “10,000 meters” with “10,000 m”

L 56. Consider replacing “2,100 meters” with “2,100 m”

L 65. Consider replacing “high temperature conditions” with “high temperatures”

L 213, 237. Consider replacing “Kruskal-wallis” with “Kruskal-Wallis”

L 272. Consider replacing “insights that with crustaceans” with “evidence that crustaceans”

L 290. Consider replacing “2,000 meters” with “2,000 m”

L 298. Consider replacing “acid” with “acids”

Author Response

Our responses to the reviewers’ comments are indicated in bold text below. 

Introduction

  • L 60. The authors should provide information about the tools and techniques used in studying animals from hydrothermal vents.
  • Lines 61-64: We added more specific information as reviewer requested.
  • L 62. The authors should define the term extremophiles, and you provide specific examples of organisms that thrive in extreme environments.
  • Lines 57-59, 66-67: We added more information about the definition of extremophiles and revised the sentence structure to enhance clarity.
  • L 67. The authors should mention the specific mechanical properties of the exoskeletons that have been found to be "good" compared to those of coastal crabs.
  • Line 72: We provided detailed information on the mechanical properties associated with the term "good."

Materials and Methods.

  • Figure 1. It is difficult to understand the location of the study areas as they overlap with photographs of crustaceans. The authors should merge this figure into several plates for a better presentation. They should include a coordinate grid with the map.
  • Lines 100, 102, 103: As the reviewer suggested, we separated several contents in Figure 1 and added a grid to the map.
  • L 174. The authors should describe in detail the specific protocols that were followed during the extraction of genomic DNA using the QIAamp Fast DNA Tissue Kit.
  • Lines 183-184: We followed the manufacturer’s protocol without modification, so we added the sentence by including the phrase 'following the manufacturer’s protocol' to the sentence like that: “Genomic DNA of specimens were extracted from the muscle tissue in a pereiopod using QIAamp Fast DNA Tissue Kit (Qiagen, Inc., Hilden, Germany), following the manufacturer’s protocol.”
  • The authors should specify what size is assumed for each species from Table 1.
  • Line 109: We added the definition of the size in Table 1.
  • Section 2.5 is redundant and should be deleted.
  • Lines 205-210: We have deleted that section.

Results.

  • L 204-206. The authors should delete this text.
  • Lines 212-214: We have deleted this text.
  • Figure 2. The authors should increase the font size.
  • Line 225: The font size was increased as per the reviewer's comments.
  • Section 3.4. In terms of statistical analysis, how do you ensure that the sample sizes (N1 and N2) used in t-tests are adequate to support the conclusions drawn from the study?
  • The sample size was limited due to challenges in collection. However, previous related research also used similarly small sample sizes, consistent with our study [1,2].

Discussion.

  • L 268. The authors should provide some hypotheses on how the convergent evolution observed in vent squat lobsters and vent crabs reflects their ecological needs in deep-sea habitats.
  • Lines 86-90: We appreciate the reviewer’s suggestion. We have already included relevant hypotheses addressing this comment at the end of the introduction section.
  • L 308. The authors should discuss in more detail how the greater weight loss attributed to CaCO3 decomposition in vent species correlates with their overall physiological resilience under high-stress conditions.
  • Lines 319-321: One notable characteristic of CaCO3 is its excellent thermal stability, making it valuable in various fields, such as in biomarker[3] and bioinspired materials[4]. We have incorporated this information accordingly.
  • L 333. What mechanisms might underlie the observed lack of significant differences in mechanical properties between habitats despite different environmental stresses?
  • The rate of convergent evolution may vary[5]. Therefore, despite the significant environmental differences between vent and coastal habitats, it is possible that mechanical properties have not yet had sufficient time to change in response to these distinct stresses.
  • L 340. The authors should specify how differences in the diet of vent crustaceans compared to coastal species might affect the elemental composition of their exoskeletons.
  • Lines 356-358: We provided more detailed explanations regarding the dietary differences between the two habitats.

Specific remarks.

  • L 54. Consider replacing “10,000 meters” with “10,000 m”
  • L 56. Consider replacing “2,100 meters” with “2,100 m”
  • L 65. Consider replacing “high temperature conditions” with “high temperatures”
  • L 213, 237. Consider replacing “Kruskal-wallis” with “Kruskal-Wallis”
  • L 272. Consider replacing “insights that with crustaceans” with “evidence that crustaceans”
  • L 290. Consider replacing “2,000 meters” with “2,000 m”
  • L 298. Consider replacing “acid” with “acids”

We have made all changes mentioned above as per  comment.

  1. Yao, H.; Dao, M.; Imholt, T.; Huang, J.; Wheeler, K.; Bonilla, A.; Suresh, S.; Ortiz, C. Protection Mechanisms of the Iron-Plated Armor of a Deep-Sea Hydrothermal Vent Gastropod. Proc. Natl. Acad. Sci. 2010, doi:10.1073/pnas.0912988107.
  2. Kobayashi, H.; Shimoshige, H.; Nakajima, Y.; Arai, W.; Takami, H. An Aluminum Shield Enables the Amphipod Hirondellea Gigas to Inhabit Deep-Sea Environments. PLoS One 2019, 14, e0206710, doi:10.1371/journal.pone.0206710.
  3. Thompson, S.P.; Parker, J.E.; Tang, C.C. Thermal Breakdown of Calcium Carbonate and Constraints on Its Use as a Biomarker. Icarus 2014, 229, 1–10, doi:10.1016/j.icarus.2013.10.025.
  4. Declet, A.; Reyes, E.; Suárez, O.M. Calcium Carbonate Precipitation: A Review of the Carbonate Crystallization Process and Applications in Bioinspired Composites. Rev. Adv. Mater. Sci. 2016, 44.
  5. Arbuckle, K.; Speed, M.P. Analysing Convergent Evolution: A Practical Guide to Methods. In Evolutionary Biology: Convergent Evolution, Evolution of Complex Traits, Concepts and Methods; Pontarotti, P., Ed.; Springer International Publishing: Cham, 2016; pp. 23–36 ISBN 978-3-319-41324-2.

Reviewer 2 Report

Comments and Suggestions for Authors

This study provides an intriguing look at convergent evolution of thermal resistance in deep-sea crustaceans. However, I have several primary concerns:

1 Sample Size: Using only two species within a single group limits the study’s breadth. It would strengthen the analysis if the authors included one or two additional species that independently evolved in hydrothermal vents.

2. Do any of the hydrothermal vent species have closely related counterparts living in the deep sea but outside of hydrothermal vents? These would make more suitable comparison species than coastal relatives, as differences observed may otherwise reflect lineage- or deep-sea-specific adaptations.

3. Could the authors clarify the rationale for using temperatures up to 220°C? While this is indeed an extreme temperature within the hydrothermal vent habitat, it seems unlikely that these organisms encounter or survive such high temperatures directly, suggesting limited selection pressure at this range. What is the organism's actual upper temperature limit, and how does its thermal resistance compare within that more ecologically relevant range?

Here are some minor comments:

Figure 1 caption: what combined dataset. Be specific on what genes used.

Line 143-148: Why did you use dehydrated samples (previous studies used them is not an answer). If you were going to link the characteristics of the samples, shouldn’t you test them in their normal conditions?   

It would be great if the authors could expand the discussion on convergent evolution in other study systems and traits to expand the impact of the paper. For example:

1) Li, R., Zarate, D., Avila-Magaña, V., & Li, J. (2024). Comparative transcriptomics revealed parallel evolution and innovation of photosymbiosis molecular mechanisms in a marine bivalve. Proceedings B, 291(2023), 20232408.

2) Lamichhaney, S., Card, D. C., Grayson, P., Tonini, J. F., Bravo, G. A., Näpflin, K., ... & Edwards, S. V. (2019). Integrating natural history collections and comparative genomics to study the genetic architecture of convergent evolution. Philosophical Transactions of the Royal Society B, 374(1777), 20180248.

3) Hu, Y., Wu, Q. I., Ma, S., Ma, T., Shan, L., Wang, X., ... & Wei, F. (2017). Comparative genomics reveals convergent evolution between the bamboo-eating giant and red pandas. Proceedings of the National Academy of Sciences, 114(5), 1081-1086.

Author Response

Our responses to the reviewers’ comments are indicated in bold text below. 

Primary concerns:

Sample Size: Using only two species within a single group limits the study’s breadth. It would strengthen the analysis if the authors included one or two additional species that independently evolved in hydrothermal vents.

  • In line with the reviewer's concern, we also wished to increase the number of species analyzed in this section; however, only two species of decapods with hard exoskeletons were available for analysis from the sampling site [6]. Additionally, including other hydrothermal vent areas in the study is challenging due to difficulties in obtaining samples.

Do any of the hydrothermal vent species have closely related counterparts living in the deep sea but outside of hydrothermal vents? These would make more suitable comparison species than coastal relatives, as differences observed may otherwise reflect lineage- or deep-sea-specific adaptations.

  • We also agree with the reviewer's concern and are preparing a follow-up paper on this matter.

Could the authors clarify the rationale for using temperatures up to 220°C? While this is indeed an extreme temperature within the hydrothermal vent habitat, it seems unlikely that these organisms encounter or survive such high temperatures directly, suggesting limited selection pressure at this range. What is the organism's actual upper temperature limit, and how does its thermal resistance compare within that more ecologically relevant range?

  • The temperature range for measuring thermal stability was set based on the average seawater temperature where hydrothermal vent fluids mix with the surrounding seawater, as noted in lines 165-168. The temperature was chosen because, according to ROV observations during sample collection, the crustaceans examined in this study were observed traversing the area above the hydrothermal vent noted in lines 117-119. While these crustaceans may not be exposed to high temperatures for extended periods, they may encounter brief exposure, supporting rationale of the temperature range.

Minor comments:

Figure 1 caption: what combined dataset. Be specific on what genes used.

  • Lines 100, 105: We rearranged the dataset in Figure 1. The information on specific genes was added followingthe advice.

Line 143-148: Why did you use dehydrated samples (previous studies used them is not an answer). If you were going to link the characteristics of the samples, shouldn’t you test them in their normal conditions?  

  • To analyze the exoskeleton of crustaceans, it is essential to remove moisture from the specimen to avoid deformation or lysis due to water evaporation during analysis [7]. Additionally, when embedding the sample in resin for cross-sectional analysis, any remaining moisture can interfere with proper fixation[8] or cause deformation or collapse of hydrated structures of the sample[9], potentially affecting the accuracy of the analysis.

It would be great if the authors could expand the discussion on convergent evolution in other study systems and traits to expand the impact of the paper. For example:

  • Lines 376-389: We added a new discussion section to address convergent evolution, as recommended by the reviewer, referencing findings from other studies.
  1. Li, R., Zarate, D., Avila-Magaña, V., & Li, J. (2024). Proceedings B, 291(2023), 20232408.
  2. Lamichhaney, S., Card, D. C., Grayson, P., Tonini, J. F., Bravo, G. A., Näpflin, K., ... & Edwards, S. V. (2019). Integrating natural history collections and comparative genomics to study the genetic architecture of convergent evolution. Philosophical Transactions of the Royal Society B, 374(1777), 20180248.
  3. Hu, Y., Wu, Q. I., Ma, S., Ma, T., Shan, L., Wang, X., ... & Wei, F. (2017). Comparative genomics reveals convergent evolution between the bamboo-eating giant and red pandas. Proceedings of the National Academy of Sciences, 114(5), 1081-1086.

References

  1. Suh, Y.J.; Kim, M.-S.; Kim, S.-J.; Kim, D.; Ju, S.-J. Carbon Sources and Trophic Interactions of Vent Fauna in the Onnuri Vent Field, Indian Ocean, Inferred from Stable Isotopes. Deep Sea Res. Part Oceanogr. Res. Pap. 2022, 182, 103683, doi:10.1016/j.dsr.2021.103683.
  2. Schädler, S.; Burkhardt, C.; Kappler, A. Evaluation of Electron Microscopic Sample Preparation Methods and Imaging Techniques for Characterization of Cell-Mineral Aggregates. Geomicrobiol. J. 2008, 25, 228–239, doi:10.1080/01490450802153462.
  3. Mollenhauer, H.H. Artifacts Caused by Dehydration and Epoxy Embedding in Transmission Electron Microscopy. Microsc. Res. Tech. 1993, 26, 496–512.
  4. Heiligenstein, X.; Lucas, M.S. One for All, All for One: A Close Look at In-Resin Fluorescence Protocols for CLEM. Front. Cell Dev. Biol. 2022, 10, doi:10.3389/fcell.2022.866472.

Round 2

Reviewer 2 Report

Comments and Suggestions for Authors

I believe that the authors have addressed all my comments. It is ready for publising.